# Intradermal Vaccination against Influenza with a STING-Targeted Nanoparticle Combination Adjuvant Induces Superior Cross-Protective Humoral Immunity in Swine Compared with Intranasal and Intramuscular Immunization

**DOI:** 10.3390/vaccines11111699

**Published:** 2023-11-07

**Authors:** Juan F. Hernandez-Franco, Ganesh Yadagiri, Veerupaxagouda Patil, Dina Bugybayeva, Sara Dolatyabi, Ekachai Dumkliang, Mithilesh Singh, Raksha Suresh, Fatema Akter, Jennifer Schrock, Gourapura J. Renukaradhya, Harm HogenEsch

**Affiliations:** 1Department of Comparative Pathobiology, College of Veterinary Medicine, Purdue University, West Lafayette, IN 47907, USA; jfhernan@purdue.edu; 2Center for Food Animal Health, Department of Animal Sciences, The Ohio State University, Wooster, OH 44691, USA; yadaigiri.1@osu.edu (G.Y.); vkpatil13@gmail.com (V.P.); bugybayeva.1@buckeyemail.osu.edu (D.B.); dolatyabi.1@buckeyemail.osu.edu (S.D.); singh.2085@osu.edu (M.S.); suresh.138@buckeyemail.osu.edu (R.S.); dr.fatemaakter16@gmail.com (F.A.); schrock.57@osu.edu (J.S.); 3Drug Delivery System Excellence Center (DDSEC), Department of Pharmaceutical Technology, Faculty of Pharmaceutical Sciences, Prince of Songkla University, Songkhla 90110, Thailand; ekachai.d@psu.ac.th; 4Purdue Institute of Inflammation, Immunology, and Infectious Disease, Purdue University, West Lafayette, IN 47907, USA

**Keywords:** vaccines, adjuvants, STING, nanoparticles, intradermal, influenza, swine

## Abstract

The development of cross-protective vaccines against the zoonotic swine influenza A virus (swIAV), a potential pandemic-causing agent, continues to be an urgent global health concern. Commercially available vaccines provide suboptimal cross-protection against circulating subtypes of swIAV, which can lead to worldwide economic losses and poor zoonosis deterrence. The limited efficacy of current swIAV vaccines demands innovative strategies for the development of next-generation vaccines. Considering that intramuscular injection is the standard route of vaccine administration in both human and veterinary medicine, the exploration of alternative strategies, such as intradermal vaccination, presents a promising avenue for vaccinology. This investigation demonstrates the first evaluation of a direct comparison between a commercially available multivalent swIAV vaccine and monovalent whole inactivated H1N2 swine influenza vaccine, delivered by intradermal, intranasal, and intramuscular routes. The monovalent vaccines were adjuvanted with NanoST, a cationic phytoglycogen-based nanoparticle that is combined with the STING agonist ADU-S100. Upon heterologous challenge, intradermal vaccination generated a stronger cross-reactive nasal and serum antibody response in pigs compared with intranasal and intramuscular vaccination. Antibodies induced by intradermal immunization also had higher avidity compared with the other routes of vaccination. Bone marrow from intradermally and intramuscularly immunized pigs had both IgG and IgA virus-specific antibody-secreting cells. These studies reveal that NanoST is a promising adjuvant system for the intradermal administration of STING-targeted influenza vaccines.

## 1. Introduction

Influenza is globally one of the leading causes of death from vaccine-preventable viral respiratory infections. Despite enormous efforts to develop an effective influenza vaccine, only suboptimal influenza vaccines have been developed with no significant progress over the past decades [1]. The continuous genetic changes in influenza viruses through reassortment and mutations make it difficult to design effective vaccines, and their ability to infect humans, pigs, birds, and other species enables the emergence of novel pandemic strains like the 2009 H1N1 pandemic influenza virus (H1N1pdm09) [2]. Similar expression of sialic acid receptors that influenza viruses exploit to establish infection in the respiratory tract in humans and pigs allows for human-to-swine and swine-to-human transmission [3,4,5]. This highlights the role of pigs as intermediate hosts for influenza viruses, making them a prime target for vaccination to disrupt the cross-species transmission that can lead to the emergence of human influenza viruses with pandemic potential.

Vaccination continues to be the most effective method for protecting pigs against the H1N1, H3N2, and H1N2 influenza A virus subtypes that are currently circulating worldwide [6]. Most vaccines employed for the control of swine influenza A virus (swIAV) in the USA are administered intramuscularly and contain three or more whole inactivated swIAV strains and subtypes [7]. The dynamic nature of influenza viruses’ antigenicity can limit vaccine effectiveness due to mismatches between vaccine strains and circulating variants, leading to diminished influenza protection in swine and humans [8,9]. Moreover, maternally derived antibodies (MDA) can interfere with vaccine-mediated responses and contribute to vaccine-associated enhanced respiratory disease (VAERD) when cross-reactive antibodies are unable to neutralize mismatched influenza strains [10,11]. Additionally, whole inactivated swine influenza vaccines containing certain types of adjuvants may induce non-neutralizing antibodies that lead to VAERD [12]. Alternative routes of vaccination such as intranasal immunization with experimental live attenuated influenza virus vaccines can provide protection against antigenically distinct variants of influenza, even in the presence of maternally derived antibodies, without eliciting VAERD [13,14]. Unfortunately, the deployment of live attenuated influenza vaccines has been discontinued because the vaccine viruses can undergo reassortment with enzootic subtypes of influenza circulating in swine [15]. Therefore, it is imperative to continue the development of adjuvanted inactivated swine influenza vaccines that are not only safe and suitable for alternative administration routes, but also effective in eliciting cross-reactive humoral and cell-mediated immunity without the risk of VAERD.

This study evaluated the efficacy of intradermal (ID), intranasal (IN), and intramuscular (IM) routes of administration for delivery of a whole inactivated influenza A virus H1N2-OH10 (WIV) vaccine containing a combination adjuvant, NanoST, comprised of a plant-derived α-D-glucan nanoparticle termed Nano-11 [16] and ADU-S100, a STING (Stimulator of Interferon Genes) agonist, against challenges with heterologous 2009 H1N1 pandemic influenza A virus (H1N1pdm09). Nano-11 offers a viable platform for developing next-generation swine vaccines, as it is prepared from a widely available resource and has been shown to be safe and suitable for various routes of vaccine delivery [17,18,19,20]. ADU-S100 is a stable derivative of naturally occurring cyclic dinucleotides (CDNs) that activate STING leading to the production of type I interferons and activation of the NF-κB signaling pathway [21,22]. Previous studies have shown that the inclusion of CDNs in vaccine formulations enhances antiviral immune responses and induces cross-protective antibody responses [23,24,25].

## 2. Materials and Methods

### 2.1. Vaccine Preparation

Swine influenza A virus H1N2-OH10 (A/Swine/OH/FAH1-10) [9] was propagated in mycoplasma-free Madin-Darby canine kidney epithelial cells (MDCK, CRL-2285, ATCC, Manassas, VA, USA) [26]. In brief, the MDCK cells were maintained in Dulbecco’s modified eagle medium (DMEM) supplemented with Antibiotic-Antimycotic (Gibco, Thermo Fisher, Waltham, MA, USA) and 10% FBS. The culture media of swIAV H1N2-OH10-infected MDCK cells was filtered with a Pellicon-2 microfiltration cassette (Millipore, Burlington, MA, USA) followed by sucrose cushion ultracentrifugation at 107,000× *g* for 4 h without breaks. The influenza virus pellet was resuspended in sterile PBS containing protease inhibitors (Sigma, St. Louis, MO, USA) and stored at −80 °C. The virus titration was conducted using serum-free DMEM with TPCK (trypsin purified from bovine pancreas, treated with L-(tosylamido-2-phenyl) ethyl chloromethyl ketone). A 0.1 M binary ethylenimine (BEI) solution was prepared by dissolving 20.48 mg of bromoethylamine (Sigma, St. Louis, MO, USA) into 1 mL of 0.2 N NaOH. Inactivation of swIAV H1N2-OH10 was achieved by treating the virus with 10 mM BEI for 6 h at 37 °C. Residual BEI was neutralized with 10 mM sodium thiosulphate (Sigma, St. Louis, MO, USA) for 2 h at 37 °C. Inactivation of the influenza was verified in MDCK cells, and the quantity of protein in the inactivated influenza pellet was determined using a micro-BCA protein assay kit (Thermo Fisher, Waltham, MA, USA).

Nano-11 was produced according to the preceding description [16]. In brief, phytoglycogen (PG) nanoparticles from sweet corn containing the *sugary-1* mutant gene were reacted with octenyl succinic anhydride (OS) and (3-chloro-2-hydroxypropyl)-trimethylammonium chloride (CHPTAC) to synthesize PG-OS-CHPTAC (Nano-11). The synthetic STING agonist ADU-S100 (Chemietek, Indianapolis, IN, USA) was electrostatically adsorbed to Nano-11. The adsorption of ADU-S100 onto Nano-11 was determined by ultraperformance liquid chromatography/tandem mass spectrometry [18]. The IN and IM vaccines were developed by combining 1 mg Nano-11 with 50 μg ADU-S100 for 1 h at room temperature to create the combination adjuvant NanoST, with 225 μg of inactivated swIAV-H1N2-OH10 subsequently adsorbed onto NanoST for an additional hour. The adjuvant dosage in ID vaccines (0.5 mg Nano-11 with 25 μg ADU-S100) was halved from that in IN and IM vaccines. Each vaccine dose contained 1 × 10^7^ TCID_50_ of swIAV before BEI inactivation and approximately 5000 HA units estimated after inactivation.

### 2.2. Animals

Specific pathogen-free (SPF) Large White–Duroc crossbred piglets were born by cesarean delivery and kept in an influenza-free BSL-2 facility. The ID vaccines were administered at 0.1 mL per dose to five-week-old SPF pigs using the needle-free PharmaJet Tropis jet injector (PharmaJet, Golden, CO, USA). Pigs were vaccinated intranasally at 0.5 mL per dose using a multidose mist delivery syringe (Prima Tech USA, Kenansville, NC, USA). Vaccines were injected intramuscularly at a volume of 0.5 mL each dose into the biceps-femoris muscle. A control group of pigs received the commercially available FluSure XP^®^ inactivated H1N1, H1N2, and H3N2 multivalent swine influenza vaccine (Zoetis, Parsippany, NJ, USA) or PBS by intramuscular (IM) injection. All pigs were immunized again 21 days later with their respective vaccine. Pigs were challenged with the heterologous H1N1pdm09 [27] strain (2 × 10^7^ TCID_50_) in 2 mL, which was administered uniformly by both intranasal and intratracheal delivery. Clinical signs were monitored two times daily and rectal temperature was documented on alternate days from the day of challenge until euthanasia. We did not observe any significant clinical signs and the pigs did not have fever. At 2, 4, and 6 days post-challenge (DPC), 2 mL of DMEM supplemented with Antibiotic-Antimycotic (Thermo Fisher, Waltham, MA, USA) was used to collect nasal swab samples. At DPC 6, pigs were euthanized and tissues were collected aseptically. Blood was collected and plasma aliquots were kept at −80 °C. Bronchoalveolar lavage (BAL) fluid was collected by infusing 40 mL of PBS containing 2% EDTA via the trachea and collecting the fluid after mild agitation of the lung lobes. Lung lysate was obtained by homogenizing 1 g of right apical lobe lung tissue in 3 mL of DMEM supplemented with protease inhibitor and centrifuging the resulting suspension to collect the supernatant. All processed tissue sample aliquots were kept at −80 °C.

### 2.3. Cross-Reactive Antibody Analysis 

IgG and secretory IgA (SIgA) antibody titers against swine influenza A virus subtypes H1N1-OH7 (A/Swine/OH/24366/2007), H1N2-OH10 (A/Swine/OH/FAH10-1/10), and H3N2-OH4 (A/Turkey/OH/313053/2004) were measured through the use of an enzyme-linked immunosorbent assay (ELISA) as described previously [18,28]. In short, semi purified swIAV H1N1-OH7, H1N2-OH10, and H3N2-OH4 viral antigens prepared by sucrose cushion ultracentrifugation were inactivated and diluted to 10 μg/mL (pretitrated concentration for ELISA) prior to the coating of 96-well plates at 4 °C for 24 h. The plates were blocked at room temperature (RT) for 2 h with 1% BSA in PBS containing 0.05% TWEEN (PBST; Sigma, St. Louis, MO, USA) after being washed three times with PBST. Serially diluted serum, nasal swab, lung lysate, and BAL fluid samples were added to the wells in duplicate for 1 h at 37 °C, followed by well washing and incubation with 100 μL of peroxidase-conjugated rabbit anti-pig IgG (A5670; Sigma-Aldrich, St. Louis, MO, USA) or HRP-conjugated goat anti-pig IgA (A100-102P; Bethyl Laboratories, Inc., Montgomery, TX, USA) for 1 h at 37 °C. All wells were treated with 100 μL of 3,3′,5,5′ tetramethylbenzidine substrate solution (Neogen, Lansing, MI, USA) after a final wash and were allowed to react in the dark for 10 min at room temperature. Absorbance at 450 nm (OD 450) was measured using a microplate reader after terminating the reaction with 50 μL of 2 M sulfuric acid. The final OD values were derived by deducting the mean of triplicate blank samples from the OD of experimental test samples. To assess antibody avidity, two sets of duplicate samples were prepared for each appropriately diluted serum sample. Following incubation with serum, one set of samples was treated with 7 M urea (Sigma, St. Louis, MO, USA), while the other set was treated with PBST (control) for 10 min. Samples were then subjected to five washes with PBST and subsequently incubated with the secondary antibody. The avidity index was calculated as OD_450_ (urea-treated samples)/OD_450_ (PBST-treated samples).

### 2.4. Influenza-Specific Antibody-Secreting Cell Analysis

Bone marrow was collected from the proximal tibias of pigs using bone marrow aspiration needles (A152; Lee Medical, Montgomery, NJ, USA) to determine the number of influenza-specific antibody-secreting cells (ASCs) by utilizing the ELISPOT assay. The day prior to euthanasia, membranes of MultiScreen IP filter plates (MAIPS4510; Millipore Sigma, Burlington, MA, USA) were activated using 20 μL/well of 35% ethanol for 30 sec and then washed with 300 μL/well of sterile PBS three times. The wells were coated overnight at 4 °C with 10 μg/mL (100 μL/well) of inactivated heterologous H1N1-OH7 or homologous H1N2-OH10 viral antigens in sterile PBS. The day of euthanasia, the plates were washed three times with sterile PBS, followed by blocking with complete RPMI (RPMI 1640 with 100 μg/mL streptomycin, 0.25 μg/mL amphotericin B, 100 U/mL penicillin, 50 mM 2-ME, 25 mM HEPES, and 2 mM L-glutamine) supplemented with 10% FBS, with samples then kept at 37 °C for 2 h. Subsequently, single-cell suspensions from bone marrow were serially diluted and incubated in duplicate for 24 h at 37 °C in 5% CO_2_. The following day, the plate was washed five times with PBST and incubated for 2 h at room temperature with biotin-labeled mouse anti-pig IgG (clone F0071241; BD Biosciences, Franklin Lakes, NJ, USA) or HRP-conjugated goat anti-pig IgA in 1% FBS-containing PBST. The plate containing biotin-labeled anti-pig IgG was washed and incubated at 37 °C for 1 h with streptavidin–HRP conjugate (ab7403; Cambridge, UK) in 1% FBS-containing PBST. Plates were treated with 3-amino-9-ethylcarbazole (Sigma-Aldrich, St. Louis, MO, USA) in the dark for 10 min to initiate enzymatic activity and were then washed 15 times with deionized water. The red-colored spots were enumerated with an ELISPOT reader (AID Diagnostika, Straßberg, Germany).

### 2.5. Preparation of Erythrocyte Suspension and Antigen

A 1% and 10% turkey erythrocyte suspension was prepared from the blood of five SPF turkeys. The erythrocytes were washed by adding 10 mL of whole blood and 35 mL of sterile PBS to a 50 mL centrifuge tube. The tube was gently inverted five times to wash the erythrocytes and then centrifuged at 800× *g* for 5 min at 25 °C. The supernatant, along with the layer of white blood cells, was carefully aspirated and discarded. The erythrocyte wash was repeated three times or until the supernatant became clear. During the last wash cycle, the supernatant was aspirated and disposed of without agitating the cell pellet. To prepare a 1% and 10% erythrocyte suspension, 1 or 10 mL of erythrocytes was added into 99 or 90 mL of sterile PBS, respectively. Serum samples were treated to remove non-specific inhibitors and agglutinins through incubation for 30 min in a 56 °C water bath, followed by mixing equal volumes of serum and 10% turkey erythrocytes suspension for 30 min at 20 °C. The mix was centrifuged at 800 × *g* for 10 min and treated sera were collected while taking care to avoid disturbing the pellet. The sera were stored in aliquots at −20 °C until ready for the HI assay. The swine influenza A virus H1N2 was inactivated with 10 mM BEI, and antigen was then derived following the steps described above in the vaccine preparation method. We established the hemagglutination (HA) units of swIAV H1N2 to create a virus dilution of 8 HA units per 50 μL to be utilized as the standard for the HI assay. 

### 2.6. Hemagglutination Inhibition (HI) Titers

The HI titers were determined as described previously [29]. In brief, an initial 25 μL two-fold serial dilution of the sera samples from 1:2 to 1:1024 was carried out in duplicates in a U-bottom 96-well plate with 25 μL sterile PBS. The wells were treated with 25 μL of 8 HAU/50 μL. Standardization included a positive control with the reference treated sera, a two-fold dilution of 8 HAU/50 μL for back titration, and a control of 1% erythrocyte suspension in sterile PBS. The plate was mixed gently by tapping the sides and then incubated at room temperature for 30 min. Subsequently, all wells received 50 μL of 1% turkey erythrocyte suspension, were mixed gently, and then incubated at room temperature for 20 to 30 min. Results were recorded when the erythrocytes control formed a distinct button, which was followed by a 45-degree angle tilt to record the presence or absence of a tear-shaped stream of the erythrocytes at the same rate as the control wells containing erythrocytes alone. The highest HI titers were presented in the log2 scale.

### 2.7. Quantification of Viral Load

Virus titers were determined as previously described [30]. Briefly, MDCK cells were seeded at a concentration of 2 × 10^4^ in 200 μL of DMEM-enriched media per well in a 96-well tissue culture plate for 24 h at 37 °C with 5% CO_2_. In tandem, sterile 96-well round-bottom plates were utilized to prepare ten-fold serial dilutions of nasal swab samples in serum-free DMEM. The confluent monolayer of MDCK cells received three washes with sterile PBS and was subsequently inoculated with 100 μL/well of the diluted nasal swab test samples. The inoculation process was carried out for a duration of 1.5 h at 37 °C with 5% CO_2_. Subsequently, to initiate and quantify virus-mediated cytopathic effects, 100 μL/well of DMEM serum-free medium supplemented with 2 μg/mL of TPCK-trypsin was added to each well and incubated for 36 h at 37 °C with 5% CO_2_. Furthermore, the cells were treated with fixation using an 80% acetone solution for a duration of 10 min and washed three times with PBS. Immunostaining was performed by incubating the cells with 50 μL/well of a 1:5000 dilution of mouse anti-influenza A virus nucleoprotein-specific monoclonal antibody (M058; CalBioreagents, San Mateo, CA, USA). This was followed by three washes with PBS and incubation with 50 μL/well of an Alexa Fluor 488-conjugated goat anti-mouse IgG (H + L) antibody (A11029; Invitrogen Life Technologies, Carlsbad, CA, USA) for 1.5 h at 5% CO_2_. A mounting media was applied consisting of a 50 μL/well solution of glycerol with a composition of 40% PBS at pH 8. The influenza A virus titer was determined by the Reed and Muench method, utilizing fluorescent microscopy (Olympus, Shinjuku, Tokyo, Japan) to record and quantify the IAV-mediated infection.

### 2.8. Virus Neutralization Titers

Virus-neutralizing (VN) antibody titers were determined as previously described [31]. Briefly, sera were heat inactivated for 30 min at 56 °C before testing. A two-fold serial dilution (starting from 1:10) of the sera was made in DMEM supplemented with Antibiotic-Antimycotic and 1% FBS in a 96-well plate. An equal volume of influenza virus (100 TCID_50_) was added to each well and allowed to incubate at 37 °C at 5% CO_2_ for 1 h. The process continued with the transfer of the sera and virus mixture into a flat-bottom 96-well plate containing a confluent monolayer of MDCK cells followed by incubation at 37 °C at 5% CO_2_ for 24 h. Subsequently, the samples were incubated with anti-swIAV nucleoprotein-specific mAb (M058, Cal Bioreagents, San Mateo, CA, USA) at 37 °C at 5% CO_2_ for 45 min, followed by Alexa-488-conjugated anti-mouse IgG (H + L) secondary antibody at 37 °C at 5% CO_2_ for 45 min. For the immunofluorescence detection of influenza virus-infected MDCK cells, the samples were observed in a 6:4 glycerol/PBS mixture.

### 2.9. Statistical Analysis

The identification of statistically significant differences for IgG or IgA antibody responses between experimental groups was determined by a two-way analysis of variance (ANOVA) test with a Tukey’s multiple comparisons test. A one-way ANOVA test followed by a Tukey’s multiple comparison test was conducted to assess statistical significance in the ELISpot, HI, and VN assays. Viral titers were analyzed by Kruskal–Wallis one-way ANOVA and Dunn’s post hoc test. GraphPad Prism (version 9.2, San Diego, CA, USA) was utilized in all statistical analyses. The results are presented as the mean value of 6–8 pigs ± SEM. A value of *p* < 0.05 indicates statistical significance.

## 3. Results

### 3.1. Intradermal Vaccination with WIV/NanoST Induces Systemic and Mucosal Cross-Reactive Antibodies

We investigated the immunological efficacy of the monovalent whole inactivated H1N2 swIAV (WIV) vaccine adjuvanted with NanoST in SPF pigs through ID, IN, or IM administration. The mock control group was injected IM with PBS and the positive control group was injected with the commercially available multivalent vaccine FluSure XP^®^ (Figure 1a). Pigs were challenged 14 days after the booster with the heterologous H1N1pdm09 virus. All vaccinated pigs demonstrated significant production of cross-reactive serum IgG against the H1N1-OH7 (heterologous), H1N2-OH10 (homologous), and H3N2-OH4 (heterosubtypic) variants when compared to the control (Figure 1b). Interestingly, pigs vaccinated ID with WIV/NanoST had the highest levels of cross-reactive antibodies. The levels of IgG in the lung lysate of pigs followed a similar pattern to those of serum IgG; however, IM vaccination with WIV/NanoST produced a slightly greater serum IgG response against H3N2 than ID immunization (Figure 1c). The IN vaccinated pigs had the lowest levels of IgG in lung lysates. Furthermore, ID vaccinated pigs had the highest IgG titers in the BAL fluid, which were significantly higher than the mock- and FluSure XP^®^-vaccinated pigs (Figure 1d). The FluSure XP^®^ vaccine did not induce significant levels of H1N2- and H3N2-specific IgG in the BAL fluid compared with the mock group. Although it is a multivalent vaccine, FluSure XP^®^ only induced a limited response. In contrast, ID and IM administration of the monovalent WIV/NanoST vaccine induced a robust cross-reactive systemic IgG response.

### 3.2. Intradermal Vaccination with WIV/NanoST Induces a Potent Cross-Reactive Mucosal IgA Response in the Respiratory System of Challenged Pigs

We next evaluated the influenza-specific IgA titers in nasal swabs, lung lysates, and BAL fluid samples from the challenged pigs (Figure 2a). All vaccinated pigs had increased concentrations of virus-specific IgA compared with the mock controls (Figure 2b). ID immunization generated the highest nasal IgA response against homologous, heterologous, and heterosubtypic viruses (Figure 2b). All vaccinated animals had increased concentrations of virus-specific IgA in lung lysate and BAL fluid reactive with the different swIAVs, except for low levels of anti-H1N2 induced by the FluSure XP^®^ vaccine (Figure 2c,d).

### 3.3. Intradermal Vaccination with WIV/NanoST Induces Cross-Reactive and Qualitative Immunological Memory

In mice and humans, the bone marrow is home to long-lived plasma cells that provide long-term immunity by actively producing antibodies [32,33]. However, previous studies in pigs did not find antigen-specific antibody-secreting cells (ASCs) in the bone marrow after infection with rotavirus or porcine reproductive and respiratory syndrome (PRRS) virus [34,35]. We first developed and optimized the procedure to detect ASCs using bone marrow from pigs immunized IN with split H1N2 virus with Nano-11 or NanoST (Appendix A). There were both IgG and IgA H1N1- and H1N2-specific ASCs in the bone marrow, with a modestly increased number in pigs vaccinated with NanoST. We assessed the presence of ASCs specific for the homologous H1N2 and heterologous H1N1 virus in bone marrow six days post challenge (Figure 3a). There were no swIAV-specific ASCs in the bone marrow of the mock-vaccinated pigs, while both IgA and IgG ASCs were detected in the vaccinated animals. The number of homologous H1N2-specific IgG and IgA ASCs was significantly increased in pigs vaccinated ID and IM with WIV/NanoST (Figure 3b). These results are consistent with serum, lung lysate, and BAL fluid IgG antibody responses (Figure 1 and Figure 2). Pigs vaccinated IM with WIV/NanoST had significantly higher H1N1-specific IgG ASCs when compared with the control group. The number of H1N1-specific IgA ASCs was significantly increased in pigs vaccinated ID or IM with WIV/NanoST and IM with FluSure XP^®^. IN vaccination induced the lowest amount of IgA ASCs against both homologous and heterologous viruses, despite the cross-reactive IgA response observed in the respiratory secretions (Figure 2). These data suggest that both the route of vaccine administration and the vaccine formulation play critical roles in the development and homing of plasma cells to the bone marrow, which may correlate with the duration of immunity.

### 3.4. Intradermal Vaccination with WIV/NanoST Induces Functional Antibodies against Influenza

The efficacy of vaccines is not only determined by the amount of antibody produced, but also by the avidity of the antibodies [36]. We therefore evaluated the effect of the vaccine and route of immunization on the avidity of the anti-swIAV serum antibody response. Pigs vaccinated ID with WIV/NanoST had the highest antibody avidity against the H1N1, H1N2, and H3N2 viruses, followed by IM vaccination with WIV/NanoST and IM vaccination with FluSure XP (Figure 4a). Pigs vaccinated via the IN route developed the lowest avidity antibody responses.

The function of anti-swIAV antibodies can be determined by hemagglutination inhibition (HI) and virus neutralization (VN) assays. The BAL fluid of pigs vaccinated ID with the monovalent WIV/NanoST vaccine had significantly higher HI titers than IN and IM immunized pigs (Figure 4b). There was no difference in BAL fluid HI titers between pigs vaccinated IM with WIV/NanoST and FluSure XP^®^. Pigs that received the FluSure XP^®^ vaccine or WIV/NanoST via the ID and IM route had significantly higher serum HI titers compared to the mock control (Figure 4c). The serum VN titers against heterologous H1N1 were increased following ID vaccination with monovalent WIV-H1N2/NanoST and IM vaccination with FluSure XP^®^ (Figure 4d).

In aggregate, these data indicate that ID vaccination with WIV/NanoST induced a superior quality of antibody response compared with IN or IM vaccination in spite of the use of a 50% dose of vaccine virus. 

### 3.5. Intradermal Vaccination with WIV/NanoST Protects against Challenges with Heterologous SwIAV

The final assessment of vaccine efficacy was the protection of vaccinated pigs against challenges with a heterologous H1N1 virus. Nasal swabs from pigs primed and boosted with PBS, WIV/NanoST, or FluSure XP^®^ were collected 2, 4, and 6 days post challenge (Figure 5). The nasal virus titer of non-vaccinated control pigs decreased from day 2 to day 6 post challenge, but virus was still present at day 6. There was no difference between the WIV/NanoST- and FluSure XP^®^-vaccinated pigs. Nonetheless, two days post challenge (DPC), the ID and IN vaccinated pigs had significantly lower viral titers compared to the mock control group. On the final day, seven out of eight pigs were negative for viral shedding in the ID vaccinated group, while all the pigs in the FluSure XP^®^-vaccinated group had no viral shedding. Thus, the WIV/NanoST vaccine inhibited the shedding of heterologous swIAV, even though the vaccine is monovalent as opposed to the multivalent FluSure XP^®^ vaccine, which contains WIV-H1N1, H1N2, and H3N2 antigens.

## 4. Discussion

Influenza A viruses are widespread in swine herds globally. Although the infections are often subclinical, they can cause significant disease outbreaks as components of the multifactorial respiratory disease complex in pigs and may cause decreased growth rate and reproductive performance, resulting in significant economic losses [6,37]. In addition, swIAVs are potential zoonotic pathogens, and infections in pigs may give rise to viruses with pandemic potential. Vaccination against influenza in swine is aimed at reducing the economic burden of influenza on the swine industry and minimizing the risk of zoonotic spillover to humans. However, the efficacy of conventional swIAV vaccines is hindered by the dynamic antigenic diversity of influenza, often rendering the vaccines ineffective at providing optimal protection. Strategies to develop more broadly protective influenza vaccines include the inclusion of multiple vaccine strains or conserved antigens, administration via different routes than traditional intramuscular injections, and the use of adjuvants [38,39,40]. We report here that needle-free ID immunization with a novel nanoparticle combination adjuvant induces a mucosal and systemic antibody response and protection against challenges with a heterologous swIAV.

Intradermal delivery is an attractive and practical route of vaccination in swine. It targets tissue that is rich in immune cells and can elicit a response that is more effective than IM injections. It can be performed with needle-free injectors, which avoids needle stick injuries and broken needles, eliminates the cost of disposing of used needles, and reduces the amount of stress inflicted on the animals [41,42,43]. Furthermore, needle-free vaccination eliminates the risk of disease transmission that occurs with the re-use of needles in multi-dose delivery devices, as demonstrated in an experimental setting with porcine reproductive and respiratory syndrome and African Swine Fever [44,45]. However, ID immunization can cause marked injection site reactions when used with the adjuvants present in traditional vaccines, such as mineral oil-based adjuvants and aluminum-containing adjuvants [23]. There is a need for adjuvants that are non-reactogenic and compatible with needle-free ID injections.

Combination adjuvants are composed of different immunostimulatory molecules that target different signaling pathways, resulting in synergistic activation of antigen-presenting cells and the induction of a robust immune response [46]. We prepared a combination adjuvant comprised of a cationic nanoparticle and the synthetic cyclic dinucleotide ADU-S100. Cyclic dinucleotides are hydrophilic, negatively charged molecules that induce robust humoral and cell-mediated immune responses when used as an adjuvant in experimental vaccines [47,48]. Binding of cyclic dinucleotides by STING induces activation of TBK1, which phosphorylates and activates the transcription factor IRF3 and results in the expression of type I and type III interferons [22]. In addition, STING ligation activates the NF-κB pathway, which induces secretion of TNF as well as other cytokines and chemokines. ADU-S100 binds human STING variants and mouse STING with high affinity [21]. This compound has been investigated as an anti-cancer immunotherapeutic drug in human patients and has been found to be safe [49]. The use of cyclic dinucleotides as vaccine adjuvants is limited by the rapid diffusion of these small molecules from the injection site, which diminishes local activity and causes systemic (“wasted”) inflammation. Encapsulation of CDNs in nanoparticles can target the CDNs to dendritic cells (DCs) in the draining lymph nodes and limit their systemic distribution [50,51,52,53].

Nano-11 is a plant-derived α-D-glucan nanoparticle that has been modified to endow the surface with hydrophobicity and an overall positive charge [16]. The nanoparticles activate human, mouse, and porcine antigen-presenting cells via multiple signaling pathways, including the NLRP3 inflammasome, NF-kB, and the p38 mitogen-activated protein kinase [20,54]. We reported previously that inactivated swIAV vaccines adjuvanted with Nano-11 were safe for IN administration in pigs; however, Nano-11 did not induce a significant systemic immune response, and only a modest detection of IgA was observed in the respiratory system of pigs [17]. To improve the efficacy of Nano-11-based vaccines, we evaluated a combination of Nano-11 and the TLR3 agonist poly(I:C), which synergistically activated porcine monocyte-derived DCs [54]. Intranasal administration of an inactivated H1N2-OH10 swIAV vaccine with the Nano-11+poly(I:C) combination adjuvant elicited a superior cross-reactive vaccine-mediated IgA response compared to IM injection of the multivalent FluSure XP^®^ swine influenza vaccine [18,22]. However, the serum IgG titers were significantly higher in pigs immunized with FluSure XP^®^ than in those vaccinated IN with Nano-11+poly(I:C). The induction of robust systemic immune responses is one of the challenges of IN vaccination. Intradermal vaccination can induce a systemic as well as mucosal immune response in humans, mice, and pigs [55,56]. We showed previously that cyclic-di-AMP and its synthetic analogue ADU-S100 are effectively adsorbed to Nano-11 and that the combination of these two adjuvants has synergistic effects on the immune response, which is consistent with the fact that each component activates different complementary signaling pathways in antigen-presenting cells. Intradermal or IM vaccination of pigs with a split swIAV with NanoST (a combination of Nano-11 and ADU-S100) induced a cross-protective mucosal and systemic antibody response in pigs [18]. Expanding on these results, we report here on the effect of different routes of immunization with whole inactivated H1N2 swIAV with NanoST on the immune response and compared the outcome with that obtained with the commercially available multivalent FluSure XP^®^ swine influenza vaccine.

Intradermal, IM, and IN administration of a WIV H1N2 vaccine adjuvanted with NanoST was safe and not associated with any local or systemic adverse reactions. Pigs that were ID or IM vaccinated with monovalent WIV/NanoST produced a systemic and mucosal cross-reactive antibody response against all the tested swIAV subtypes. Consistent with our previous studies in mice [20], the ID route allowed for dose-sparing as it induced an equal or greater antibody response with higher antibody avidity than IM or IN immunization with half the dose. The ID vaccine-induced response was superior to that observed in FluSure XP^®^-immunized pigs. In contrast to IM injection, ID vaccination with WIV/NanoST also induced robust mucosal IgA titers similar to those obtained with IN vaccination. This is consistent with previous studies indicating that ID immunization can induce respiratory immunity in pigs [57,58].

The bone marrow is the major site of B cell development in pigs, as it is in most other mammalian species [59]. In mice and humans, B cells can differentiate into long-lived plasma cells that return to the bone marrow and account for the long-term persistence of serum antibody titers following infection and vaccination [32,60]. Previous studies in swine reported few if any antibody-secreting cells (ASCs) in the bone marrow following infection with rotavirus and PRRS virus [34,35]. Quantification of plasma cells by flow cytometry based on the expression of Blimp1 and IRF-4 revealed relatively few cells, leading the authors to suggest that the bone marrow plays a different role in plasma cell biology in pigs compared with humans and mice. In contrast to these reports, we found a substantial number of virus-specific ASCs in the bone marrow of pigs following vaccination with WIV with NanoST and with the commercial FluSure XP^®^ vaccine. Vaccination induced both IgG and IgA swIAV-specific ASCs regardless of the route of immunization. Studies in mice have shown that the bone marrow is the site for long-lived IgG and IgA plasma cells [61]. Whether the ASCs in the bone marrow of pigs after vaccination with WIV represent true long-lived plasma cells remains to be determined; however, our studies for the first time demonstrate the induction of ASCs in the bone marrow following vaccination in independent experiments (Figure 3 and Appendix A). The reason for the discrepancy with previous studies in which ASCs were not detected is unclear but may be related to the method of exposure, namely oral or intramuscular infection with a live virus [34,35] versus vaccination with an inactivated virus with a strong adjuvant. Adjuvants are thought to contribute to the development of long-lived plasma cells following vaccination, although the underlying mechanisms are not well understood [62].

Pigs vaccinated IN with WIV/NanoST generated a relatively weak antibody response, which correlates with the low numbers of ASCs in the bone marrow. This is likely due to the pig’s immunological tolerance in the respiratory tract. Respiratory viruses may not be completely eradicated by the immune system during natural infections due to the evolutionary tolerance of the immune system towards respiratory agents [63,64]. As a result, the immune system may allow for transient respiratory infection by viruses like influenza to avoid activating the full destructive capabilities of the immune system, which therefore limits the development of long-lasting immunity [65]. Nevertheless, IN vaccination offers an advantage over parenteral routes of immunization as it is less susceptible to interference by maternal antibodies [14,66]. Given the propensity of ID vaccination to induce both systemic and mucosal immune responses, a combination of IN priming and ID booster vaccination may induce the most effective immune response to protect against respiratory pathogens.

In conclusion, our investigation demonstrates the potential of ID vaccination with the NanoST combination adjuvant to induce cross-protective systemic and mucosal immunity against diverse genetic variants of swIAV. The needle-free ID route of immunization was safe and induced a superior immune response with high avidity IgA and IgG antibodies. We also report for the first time that vaccination induced IgA and IgG ASCs in the bone marrow of swine, which may underlie a durable immune response. Our findings further suggest that the route of immunization be considered for the development of next-generation vaccines for animals and humans.

## Figures and Tables

**Figure 1 vaccines-11-01699-f001:**
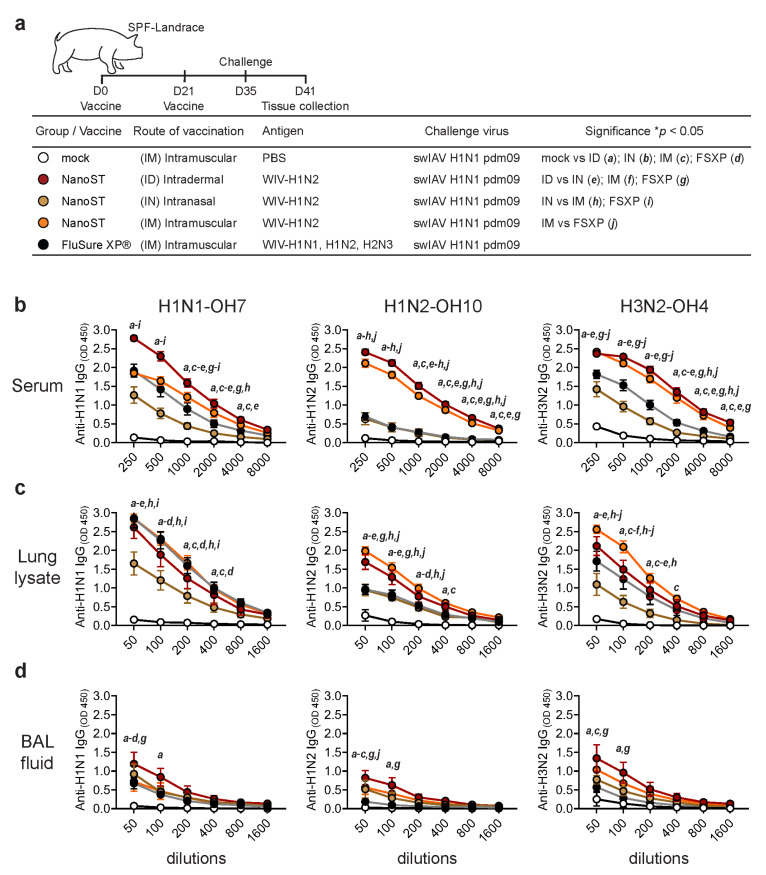
Cross-reactive influenza-specific IgG in the serum and respiratory tract of pigs after vaccination with WIV/NanoST. (**a**) SPF pigs were primed and boosted 21 days apart with the monovalent whole inactivated H1N2 virus adjuvanted with NanoST (WIV/NanoST) by intradermal (ID), intranasal (IN), or intramuscular (IM) administration. To serve as a comparison group, pigs were IM injected with either the commercially available multivalent FluSure XP^®^ influenza vaccine or PBS. All pigs were challenged with the pandemic H1N1 virus (H1N1pdm09) 14 days after the booster and were euthanized 6 days post challenge. Samples of serum (**b**), lung lysate (**c**), and BAL fluid (**d**) were analyzed by ELISA for the identification of influenza-specific IgG responses against H1N1-OH7 (heterologous), H1N2-OH10 (homologous), and H3N2-OH4 (heterosubtypic) subtypes. The data represent the mean ± SEM of 6–8 animals per group. * *p* < 0.05 as determined by a two-way ANOVA with Tukey’s multiple comparison test.

**Figure 2 vaccines-11-01699-f002:**
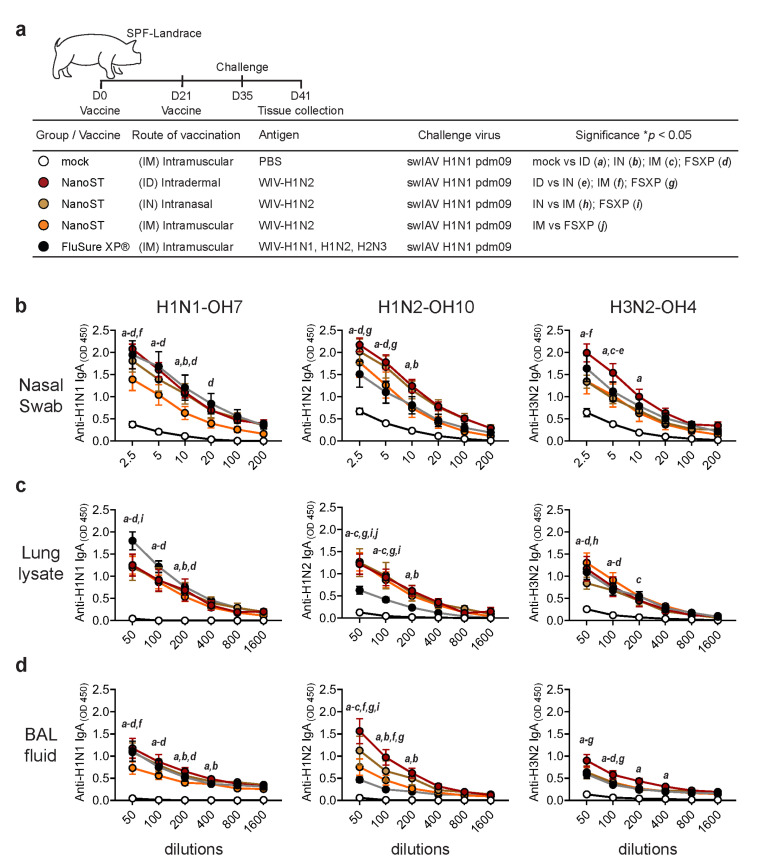
Vaccination of pigs with WIV/NanoST induces the production of cross-reactive influenza-specific secretory IgA. (**a**) SPF pigs were vaccinated twice 21 days apart via intradermal (ID), intranasal (IN), or intramuscular (IM) delivery with WIV/NanoST. In a control group, pigs were injected IM with either the commercially available FluSure XP^®^ influenza vaccine or PBS. All pigs were challenged with the H1N1pdm09 virus 14 days after the second vaccination and were euthanized 6 days following the challenge. ELISA tests were used to determine influenza-specific IgA responses against H1N1-OH7 (heterologous), H1N2-OH10 (homologous), and H3N2-OH4 (heterosubtypic) subtypes in nasal swab (**b**), lung lysate (**c**), and BAL fluid (**d**) collections. Results are presented as the mean ± SEM of 6–8 pigs. * *p* < 0.05 by a two-way ANOVA with Tukey’s multiple comparison test.

**Figure 3 vaccines-11-01699-f003:**
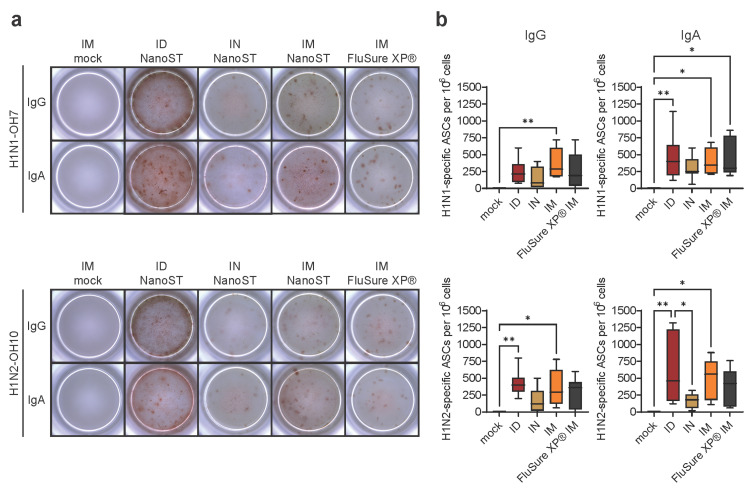
Vaccination with WIV/NanoST stimulates the generation of cross-reactive bone marrow-derived antibody secreting cells (ASCs). Bone marrow-derived ASCs were evaluated for IgG and IgA reactivity to H1N1-OH7 (heterologous) and H1N2-OH10 (homologous) viral antigens utilizing ELISpot. (**a**) A representative image from one pig for each experimental group, viral antigen, and vaccine administration route is shown. (**b**) Results are presented as influenza-specific ASCs/per 10^6^ bone marrow-derived cells. Results are presented as a box and whisker plot of 6–8 pigs. * *p* < 0.05, ** *p* < 0.01 by one-way ANOVA with Tukey’s multiple comparison test.

**Figure 4 vaccines-11-01699-f004:**
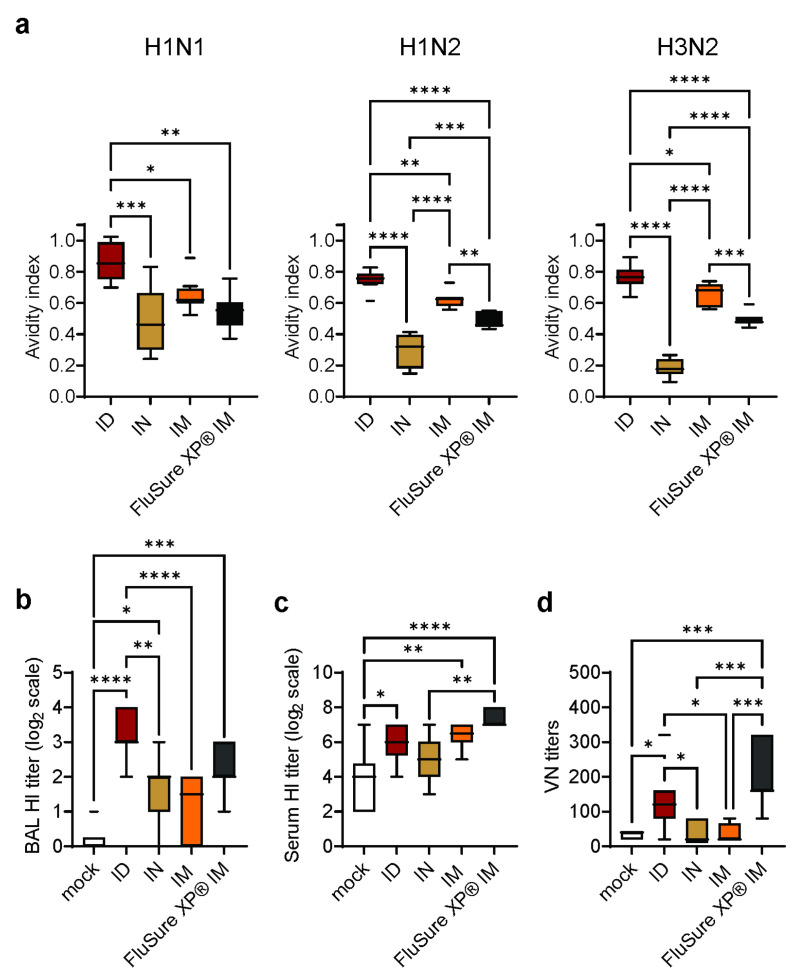
NanoST induces a high-quality antibody response with the functional capacity of viral neutralization. SPF pigs were vaccinated twice with monovalent WIV-H1N2/NanoST by intradermal (ID), intranasal (IN), or intramuscular (IM) administration and challenged 14 days after the second vaccination with H1N1pdm09. The mock control group was vaccinated with PBS via the IM route, and the positive control group received the commercially available multivalent FluSure XP^®^ via the IM route. Serum and bronchoalveolar lavage (BAL) fluid samples were collected at six days post challenge. (**a**) Avidity of H1N1-, H1N2-, or H3N2-specific serum IgG. Hemagglutination inhibition (HI) antibody titers in the BAL fluid (**b**) and serum (**c**) are presented in the log2 scale. (**d**) Serum virus neutralization (VN) titers were quantified by the cell culture technique and immunofluorescence. Data represent the mean value of 6 to 8 pigs ± SEM. * *p* < 0.05, ** *p* < 0.01, *** *p* < 0.001, **** *p* < 0.0001 by one-way ANOVA with Tukey’s multiple comparison test.

**Figure 5 vaccines-11-01699-f005:**
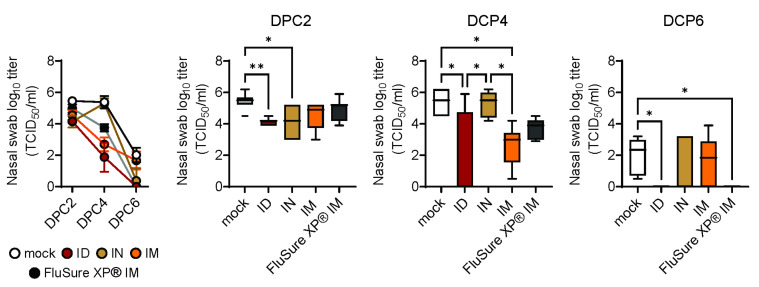
Intradermal vaccination with WIV/NanoST provides an immediate and persistent decrease in viral shedding. Virus titers at 2, 4, and 6 days post challenge (DPC) were determined by the cell culture technique and immunofluorescence. Data represent the mean ± SEM of 6 to 8 pigs. * *p* < 0.05, ** *p* < 0.01 by Kruskal–Wallis one-way ANOVA and Dunn’s post hoc test.

## Data Availability

Data will be made available by the corresponding authors upon reasonable requests.

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
