# Peer review of "Intradermal Vaccination against Influenza with a STING-Targeted Nanoparticle Combination Adjuvant Induces Superior Cross-Protective Humoral Immunity in Swine Compared with Intranasal and Intramuscular Immunization"

_vaccines, 2023, doi:10.3390/vaccines11111699_

Round 1
Reviewer 1 Report
Comments and Suggestions for Authors
This article explores different routes of vaccine administration of an inactivated H1N2-OH10 influenza virus-based vaccine in a formulation with Nano-11 nanoparticles and the STING agonist ADU-S100. The authors evaluated the effect of the experimental vaccine administered via ID, IN, and IM and its ability to stimulate IgG and IgA antibody production in serum, mucosa, avidity, HI titers, and viral neutralization. In addition, they showed the presence of antigen-specific plasma B lymphocytes in bone marrow, suggesting the induction of this "long-lived" antibody-secreting cell population upon evaluation. According to the results, the ID vaccination strategy showed better advantages in the stimulation of systemic and mucosal humoral immunity, as well as higher quality (affinity, neutralizing capacity, and inhibition of hemagglutination) in the antibodies produced against 3 homologous and heterologous variants of the influenza virus. In some cases, the experimental vaccine showed a better response compared to an inactivated multivalent vaccine FluSure XP (with H1N1, H1N2, and H3N2 variants). On the other hand, the ID immunization strategy also proved to be efficient in rapid viral load clearance after pigs were challenged with the H1N1pdm-09 variant. However, there are a few minor details that need to be addressed.
- In the introduction, the rationale for exploring routes of vaccination other than intramuscular and applying appropriate adjuvants other than those used by the IM route is well justified. However, the approach of choosing the NanoST system composed of Nano-11 nanoparticles and the STING agonist ADU-S100 is not fully supported.
· Line 161-162: Why did you not evaluate the antibody-secreting cells specific to the heterosubtypic H3N2-OH4 in ELISPOT assays?
· Figure 1 and 2. The representation of statistical differences between groups is confusing.
· Line 291: “…and heterosubtypic viruses. (Fig. 2b).” There is an extra period in the sentence.
· Line 323: A period seems missing after “…WIV/NanoST and IM with FluSure XP®”.
· Lines 325-327: “These data suggest that the route of vaccine administration plays a critical role in the development and homing of plasma cells to the bone marrow which may correlate with the duration of immunity”. Pigs vaccinated I.M. and I.D. with WIV/NanoST vaccine showed higher H1N1 y H1N2-specific IgA and IgG ASC in bone marrow compared with other groups, even with the FluSure XP I.M. vaccinated group. It seems contradictory to suggest that the vaccination route plays a critical role in developing and homing plasma cells to the bone marrow since FluSure XP vaccine was applied I.M. and did not stimulate an effect similar to the I.M. applied WIV/NanoST vaccine.
· Line 428: Since “CDNs” is mentioned for the first time it would be convenient to Include the meaning.
· Why the information on the clinical signs of the pigs after infection was not included?
· What would be why the monovalent vaccine seems more effective in stimulating cross-protection against variants of SwIA even compared to the multivalent vaccine?
Author Response
Response to the comments from Reviewer 1.
We thank the reviewer for the constructive criticisms and suggestions.
• In the introduction, the rationale for exploring routes of vaccination other than intramuscular and applying appropriate adjuvants other than those used by the IM route is well justified. However, the approach of choosing the NanoST system composed of Nano-11 nanoparticles and the STING agonist ADU-S100 is not fully supported.
RESPONSE: Thank you for this suggestion. We have added text to explain the choice of STING agonists at the end of the Introduction (lines 78-82).
• Line 161-162: Why did you not evaluate the antibody-secreting cells specific to the heterosubtypic H3N2-OH4 in ELISPOT assays?
RESPONSE: We initially planned to focus on the cross-reactivity between the H1N2 vaccine virus and the H1N1 challenge virus. We decided to include H3N2 in the analysis of the antibody responses at a later time, after the ELISpot assay for H1N1 and H1N2 had already been completed.
• Figure 1 and 2. The representation of statistical differences between groups is confusing.
RESPONSE: We acknowledge that the multiple comparisons between the five experimental groups can be somewhat time-consuming to evaluate. However, we believe that the graphs allow for a straightforward comparison between the antibody binding to different SwIAV strains and the different sources (serum, nasal, etc.) of antibodies.
• Line 291: “…and heterosubtypic viruses. (Fig. 2b).” There is an extra period in the sentence.
RESPONSE: Thank you. The period has been removed (line 297 in the revised manuscript).
• Line 323: A period seems missing after “…WIV/NanoST and IM with FluSure XP®”.
RESPONSE: Thank you. The period has been added (line 329 in the revised manuscript).
• Lines 325-327: “These data suggest that the route of vaccine administration plays a critical role in the development and homing of plasma cells to the bone marrow which may correlate with the duration of immunity”. Pigs vaccinated I.M. and I.D. with WIV/NanoST vaccine showed higher H1N1 y H1N2-specific IgA and IgG ASC in bone marrow compared with other groups, even with the FluSure XP I.M. vaccinated group. It seems contradictory to suggest that the vaccination
route plays a critical role in developing and homing plasma cells to the bone marrow since FluSure XP vaccine was applied I.M. and did not stimulate an effect similar to the I.M. applied WIV/NanoST vaccine.
RESPONSE: That is a good point. We have changed the sentence to: “These data suggest that both the route of vaccine administration and the vaccine for-mulation play critical roles in the development and homing of plasma cells to the bone marrow which may correlate with the duration of immunity.”
(lines 331-334 in the revised manuscript).
• Line 428: Since “CDNs” is mentioned for the first time it would be convenient to Include the meaning.
RESPONSE: Thank you. We have included cyclic dinucleotides (CDNs) in the added text at the end of the Introduction (lines 78-82).
• Why the information on the clinical signs of the pigs after infection was not included?
RESPONSE: We did not observe any clinical signs in the pigs in spite of twice daily observations and measuring the body temperature. We have added a statement to the Methods section (lines 127-128).
• What would be why the monovalent vaccine seems more effective in stimulating cross-protection against variants of SwIA even compared to the multivalent vaccine?
RESPONSE: That is an interesting question. We believe that the NanoST adjuvant stimulates a robust germinal center reaction similar to what we observed in a previous study in mice using Nano11 and cyclic-di-AMP (Ref. 20). A strong and prolonged GC reaction may lead to cross-reactive antibodies. However, at this time, we don’t have data to compare the GC reactions following vaccination of pigs with NanoST vs. FluSure XP.

Reviewer 2 Report
Comments and Suggestions for Authors
The authors demonstrated that ID vaccination with the NanoST combination adjuvant induced systemic and mucosal immunity in pigs against swIAV more than IN or IM vaccination. The findings are of interest. However, they seemed to demonstrate no or little cross-reactivity of the vaccine. ELISA and ELISPOT assay seemed to indicate the reactivity with the nucleoprotein, which is common to influenza A virus. Furthermore, any data except virus recovery obtained after the challenge did not demonstrate true vaccine effect.
Minor points:
1. Key words should include pig or swine.
2. Line 43. 2009 H1N1 pandemic influenza A virus (H1N1pdm09) not 2009 H1N1 influenza virus.
3. Lines 108-130. The authors should indicate Animal Experiment Committee Approval Number.
4. Line 132-134. The strain name should be indicated.
5. Subsections 2.7 and 2.8 should be swapped.
6. Line 429. Is CDNs cyclic dinucleotides?
Comments on the Quality of English LanguageThe authors should be careful about use of abbreviations.
Author Response
We thank the reviewer for the constructive criticisms and suggestions.
The authors demonstrated that ID vaccination with the NanoST combination adjuvant induced systemic and mucosal immunity in pigs against swIAV more than IN or IM vaccination. The findings are of interest.
However, they seemed to demonstrate no or little cross-reactivity of the vaccine. ELISA and ELISPOT assay seemed to indicate the reactivity with the nucleoprotein, which is common to influenza A virus. Furthermore, any data except virus recovery obtained after the challenge did not demonstrate true
vaccine effect.
RESPONSE: We used whole inactivated virus as antigen for the ELISA and ELISpot assays. Antibodies will bind to surface proteins, mostly HA, and will not be able to access the internal nucleoprotein.
Minor points:
1. Key words should include pig or swine.
RESPONSE: Thank you for this suggestion. We have added “swine” as a keyword.
2. Line 43. 2009 H1N1 pandemic influenza A virus (H1N1pdm09) not 2009 H1N1 influenza virus.
RESPONSE: Thank you. We have made the correction (line 43).
3. Lines 108-130. The authors should indicate Animal Experiment Committee Approval Number.
RESPONSE: This is listed at the end of the manuscript according to the journal’s guidelines (lines 531-534)
4. Line 132-134. The strain name should be indicated.
RESPONSE: The strain names have been added (lines 139-140).
5. Subsections 2.7 and 2.8 should be swapped.
RESPONSE: Done.
6. Line 429. Is CDNs cyclic dinucleotides?
RESPONSE: That is correct. The abbreviation has been introduced in the revised Introduction (line 79).

Reviewer 3 Report
Comments and Suggestions for Authors
The reviewed manuscript describes the development of cross-protective vaccine against the zoonotic swine influenza A virus. The monovalent inactivated H1N2 swine influenza (swIAV) vaccine adjuvanted with a cationic phytoglycogen-based nanoparticle that was combined with the agonist of stimulator of interferon genes (STING) was compared with the commercially available multivalent swIAV vaccine. Intradermal, intranasal, and intramuscular delivery of vaccines were tested. Nasal and serum antibody response was evaluated. The ELISPOT assay was used to determine the number of influenza specific antibody-secreting cells (ASCs). Both IgA and IgG ASCs were detected in the in the bone marrow of the vaccinated animals, while no swIAV-specific ASCs in the mock-vaccinated pigs. These results was consistent with serum, lung lysates and BAL fluid antibody responses. This work makes a significant contribution to the development of the zoonotic swine influenza A virus.
The work is well illustrated and written, and can be accepted in its present form.
Author Response
We thank the reviewer for taking the time to read our manuscript and for his/her positive comments.
Round 2
Reviewer 2 Report
Comments and Suggestions for Authors
The authors blocked ELISA plate with BSA in PBS containing Tween and washed three times with PBST, which might disrupt the virus. It means that the sera and ASCs reacted with the nucleoprotein, which is common to influenza A virus.
